# Methodology for Verifying the Indication Correctness of a Vessel Compass Based on the Spectral Analysis of Heading Errors and Reliability Theory

**DOI:** 10.3390/s22072530

**Published:** 2022-03-25

**Authors:** Krzysztof Jaskólski

**Affiliations:** Department of Navigation and Marine Hydrography, Faculty of Navigation and Naval Weapons, Polish Naval Academy, ul. Śmidowicza 69, 81-127 Gdynia, Poland; k.jaskolski@amw.gdynia.pl; Tel.: +48-2-6126-2837

**Keywords:** compass accuracy, compass integrity, band-stop filter, filter with a finite impulse response (FIR), amplitude spectral analysis, Fourier transform, stochastic processes, Markov chain

## Abstract

This article presents a novel method for validating compass devices. To this end, the post-processing method was used, i.e., the previously recorded vessel’s heading from three compass devices was applied. A spectral analysis of the recorded heading in the frequency domain was conducted by applying the fast Fourier transform method. Based on a synthetic summary of the results of spectral analysis in the heading error frequency domain, the factors causing errors of low-frequency compass device indications, manifested by the vessel’s yawing due to the vessel steering errors on the pre-determined course and the external factors causing total drift of the vessel, were eliminated. To this end, the convolution functions in the form of the sum of input signals with an impulse response, i.e., the filter with a finite impulse response (FIR) and with an infinite impulse response (IIR), were used to compare the effectiveness of the methods estimating the vessel’s heading. The final stage of the research process in the methodology applied was the use of state- and time-discrete Markov processes whose task is to determine the matrices of the intensity of transitions between the states of individual compass systems.

## 1. Introduction

The development of new compass designs is an ongoing process. Currently, the rate of compass development is taking on an exponential character. Modern compasses are built based on different physical phenomena. The development trend of using increasingly advanced technical solutions results from the needs and intended use of the ships. Therefore, traditional solutions in the form of classic compasses repeatedly fail to meet expectations in terms of accuracy, weight, dimensions, and other characteristics. The weight, dimensions, voltage, and power consumption of devices are critical aspects in the selection of compass equipment for unmanned vehicles.

A gyrocompass used as equipment in both surface vessels and submarines became a standard in the 20th century and is commonly used as an excellent solution on transport ships as well [1]. This technical solution is not free of defects and limitations, which include the weight, price, and the requirement of a stable power supply, which precludes the possibility of installing these devices on motorboats or drones. An additional problem in the operation of gyroscopic compasses includes sensitivity to vibrations, shocks, environmental impact, and accelerations that occur on vessels performing intense maneuvers.

The problems arising from the use of gyrocompasses have provided the impetus to replace the traditional solutions based on mechanical gyroscopes with fully electronic designs. This category of compasses includes electronic systems using the Earth’s magnetic field (fluxgate, inductive sensors, anisotropic magnetoresistors), electronic systems using the phenomena of Earth’s rotation, gravitation, or the measurement of accelerations (analytical compasses), and electronic systems using signals transmitted by navigation system satellites (satellite compasses). The latter give rise to much controversy in the maritime community. While other compasses operate based on natural physical phenomena, a satellite compass is not autonomous, and requires an artificially created signal source that is susceptible to interference caused by either unauthorized persons or selective access and the shutdown of the positioning system by the owner.

Therefore, the need to have a device that performs the task of a compass, i.e., ensures spatial orientation, is found on all moving objects including surface vessels and submarines. It is worth noting that without a compass, not only many navigation tasks, such as the steering of a moving object, but also collision avoidance tasks or position determination cannot be performed. Without this information, it is not possible to ensure the proper functioning of many shipborne devices. Even the commonly used radar with no compass device connected displays a blurry image on the screen, giving the impression of a shaky photograph. The heading from compass devices is also used to present the movement direction of objects equipped with automatic identification system (AIS) devices. In many cases, the information derived from this device is used in anti-collision systems and systems for supporting the watch officer’s decisions, e.g., the NAVDEC system [2] where the assessment of the AIS dynamic information relevance for the purposes of anti-collision systems is provided, e.g., in [3].

The understanding of the nature of compass functioning from the perspective of a person responsible for the vessel navigation safety requires an understanding of the nature of the errors of a compass as a measuring instrument, the interpretation of its indications, and the assessment of the integrity of the data presented and the manner of determining the measurement imperfections of the device. The nature of compass device errors can usually be different for each device type. The complex design of compasses indicates that some components of the system may be operating properly, yet errors may still occur. Compass manufacturers commonly use the principle of defining the measurement quality as the mean measurement error, as if it were a totally random process. Since errors in the form of the measurement indeterminacy of various devices are also characterized by a different variation spectrum, not only the amplitude of these changes is of significance but also the frequency of a particular error occurrence.

A classic gyrocompass, as well as a Fiber Optic Gyro (FOG) compass and a Ring Laser Gyro (RLG) compass are part of the equipment of conventional vessels. Commercial ships are equipped with a classic compass, while research, hydrographic, and training vessels are usually equipped with FOG and RLG-type compasses. A different category is the equipment of unconventional and conventional vessels with compasses using the technique, including the use of artificial navigation system satellites, particularly satellite compasses. The above-mentioned compasses are a source of numerous studies and experiments aimed at the elimination of operational errors and the characterization of the source of their origin [4,5,6,7]. In [4], simulations of the operation of a classical gyrocompass were presented. The purpose of this article was to present the emerging compass static and dynamic errors. This task was carried out from the moment the gyrocompass was activated, through the observation of the behavior of the main axis of the gyroscope. The articles [5,7] present a different approach and analyze compass errors in the frequency domain. In contrast to that, Ref. [6] presents the results of heading accuracy tests, in which the vehicle heading was determined based on the increments of Cartesian coordinates of a vehicle traveling on a tram track with the use of GNSS-RTK receivers.

Various technical solutions applied in compass devices will have a variety of errors that can be observed in the time domain. The gyroscopic sphere in classic gyrocompasses will maintain its position in space based on the physical pendulum principle. Consequently, oscillations of the main gyroscope axis will occur immediately after performing a vessel maneuver. In particular, the inertial deviation phenomenon will occur when the vessel is rapidly maneuvered with a heading change of 180 degrees.

Unlike classic gyrocompasses, since the FOG-type compasses and satellite compasses have no mechanical gyroscopes, the occurrence of errors of a different nature can be expected. With a few exceptions, it is difficult to find literature on studies into the accuracy and behavior of modern compasses installed on vessels under marine conditions [5,8,9,10,11,12]. The most commonly applied approach is to determine random errors defined as standard deviation. It is also worth noting that this process is not defined as a normal distribution. This process requires knowledge of the sensors’ error spectrum [5,7,8]. In article [8], the reference heading from the GPS-RTK system was used to determine the error of the gyro and magnetic compass. In contrast to [7], the reference heading was determined based on the heading estimated from three compasses.

Many scientific articles have focused on the determination of gyrocompass errors that are in the nature of a low-frequency ergodic process [13,14]. Moreover, knowledge of the nature of errors of advanced compass devices is insufficient. A few studies in this regard have been conducted by the Polish Naval Academy in Gdynia [8] using the STRAPDOWN technology based on mechanical gyroscopes.

A more detailed analysis was conducted and described in [15], while for satellite compasses, the spectral analysis of the heading error amplitude in the frequency domain is presented in [7]. The research outcomes were presented for static and dynamic measurements along the planned route. Interesting results of a study using numerical methods in the simulation and a static experiment using low-budget inertial measurement units (IMU) with FOG-type three-axis gyroscopes and accelerometers using the MEMS technology are provided in [16]. The article presents how the numerically simulated gyroscope sensitivity changes and how accuracy is affected by the measurement noise of gyroscopes and accelerometers. Interesting results were also obtained in a study involving directional gyroscopes using the MEMS technology along with an application for determining the north direction and the object’s position with a gyroscopic drift not exceeding 0.03 degrees per hour [17]. A similar accuracy analysis with a limitation from one device to conduct the tests is presented in [18,19,20]. In addition, the study results were presented on small autonomous vessels [21,22,23,24,25,26,27].

Many aspects concerning the integration of inertial technique with the GPS system have been presented in publications by [28,29]. An interesting test of the prototype GNSS/IMU compass using fiber-optic gyroscopes was presented by [30]. The presented research results confirmed the compass accuracy of ±1.5 degrees (3σ). Reference attitude parameters were obtained using the data from reference INS with heading generation accuracy of 6′ s LAT. The article demonstrates results with signals from multiple satellites, one satellite, and phase measurements generation and the results under total GNSS signal outage.

Marine accidents have often resulted from incorrect indications of compass devices coupled to the autopilot. Gyrocompass indication errors are actually minor, but for automatic vessel steering, a backup source of information on the course in the form of data derived from the second gyrocompass or a fluxgate-type compass is required. The research material collected during the measuring experiment can be used in the analysis of the effects associated with the vessel’s movement dynamics and in the analysis related to the compass indication reading accuracy. It is assumed that the main factor of the increase in the heading error amplitude is related to the vessel’s dynamic movement. The vessel’s movement determines both erroneous compass indications and the measurement error.

A comparison of the compass heading with the reference heading ensures a complete analysis of changes in the compass course resulting from the vessel’s dynamics. Such a comparison, however, provides information on the absolute values of compass errors for specified observation moments, and in no way represents the character of the occurring device errors. This analysis indicates that it is important to not only analyze the error distribution in the time domain but also in the frequency domain, based on the amplitude spectrum of the heading error frequency. One of the basic tools known from the digital signal processing theory is the Fourier transform applied in spectral analysis in the frequency domain.

This article presents a novel method for validating compass devices. The solution is based on the analysis of the heading oscillation spectrum and the reliability theory, which, based on the process of quantizing the operational state of a particular system, specifies whether, and at what point, the device is applicable in the navigation process. In order to present the details of the methodology for testing the accuracy of compass devices, the article is structured as follows. Section 2 presents the principles adopted when performing the measuring campaigns. The section discusses the adopted sampling frequency, the principle of determining the reference heading, the principle of determining the heading errors based on the difference of 3 compasses indications with the reference rate coming from the hydrographic system, the methodology for the spectral analysis of signal amplitude in the frequency domain using the fast Fourier transform (FFT) where the harmonics determine the frequency of occurrence of compass errors, the methodology for applying the convolution function as the sum of input signal samples with an impulse response, i.e., a filter with an infinite impulse response (IIR) and a filter with a finite impulse response (FIR), which was used to eliminate compass errors, where harmonic components of compass errors occur at low frequencies, and the methodology for testing compass integrity based on state- and time-discrete stochastic processes—the Markov chain. Section 3 presents the results of the tests conducted in accordance with the methodology for data processing, described in Section 2. Section 3 describes how the factors causing indication errors for low frequencies of the three compasses were eliminated based on the presented results of spectral analysis in the heading error frequency domain. In order to complete this stage of research work, a filter with an infinite impulse response and a filter with a finite impulse response along with a band-stop filter, high-pass (0–20 mHz), and band-pass (5–15 mHz) filter were used. The final stage illustrated the results of testing of the integrity of the compass device operation by determining the matrix of the intensity of transitions between operational states of individual compass systems before and after the application of the reference heading comparison methods. The overall process of validation of the three compasses is summarized in the ending and conclusions from performed research work and the nature of future research.

The article answers the following questions:−Does an FIR or IIR band-stop filter effectively eliminate low-frequency oscillations of the compass heading?−In what frequency range should the stop-band for the harmonic components be applied in the case of the compass heading amplitude oscillations caused by the vessel steering errors?−What state criterion in Markov processes should be adopted in order to determine matrices of the intensity of transitions between the states of the devices, and to detect differences in the accuracy of devices when using a signal before the application of filtering and after the application of band-stop filtering?−To what extent can the information derived from compass devices be useful as an input signal to devices for automatic steering of an object’s movement and the AIS devices?

## 2. Materials and Methods

### 2.1. The Measuring Experiment Course

A satellite compass, a classic gyrocompass, and a Fiber Optic Gyro (FOG) compass transmit NMEA 0183 messages with the header HDT (heading true) or HDG (heading gyro). When describing the study performance in detail, the definition and role of the NMEA standard in the navigation message, derived from the material recorded by a compass device, need to be explained. The NMEA-type coding standard is used to record various navigation parameters available from electronic devices installed on marine vessels in the text form. Using the knowledge of this standard coding, one can read individual data relevant to the research problem.

A test stand using the FOG NAVIGAT 3000, NAVIGAT 10 MK1, and FURUNO SC50 devices was prepared by the Department of Navigation and Marine Hydrography and installed on the hydrographic vessel. The study was conducted in the Gdańsk Bay area during the performance of marine sounding in accordance with the direction of hydrographic profiles. The state of the sea of 1–2, the wind affected the vessel from the SW-2B (Beaufort). Three measurement campaigns were conducted. The duration of a single measuring campaign was approximately 1000 s. The frequency for sampling recorded data was adopted at f(s) = 0.2 Hz. The test results for the FIR model with a stop-band (5–15 mHz) have already been presented in [5]. The added value in the presented methodology is obtaining a reference course from the POS MV OceanMaster Applanix Trimble hydrographic system and comparing it with the values recorded from 3 types of compasses. In addition, the added value will be the use of FIR and IIR filters for different stop and pass bands, taking into account gyrocompass errors occurring after 180 degree heading alteration. The accuracy of the devices used in the study and the mean error value are presented in Table 1. The mean latitude of the area in which the measurements were conducted is 54.6 degrees N. Mean error values were determined for gyrocompasses based on the mean latitude.

### 2.2. The Accuracy of a Vessel’s Heading

One of the research work results should be a new or improved method for data processing, the use of which will ensure the direction measurement with an accuracy comparable to or higher than that of standard navigation instruments. In order to fulfil the task undertaken, the following thesis was proposed: The application of convolution function in the form of an FIR and IIR filter will enable the determination of a heading that will be more accurate than (or comparable to) the heading determined using standard compasses.

The International Maritime Organization (IMO) imposes accuracy criteria on device manufacturers, the fulfilment of which will be equivalent to the certification of the instrument for use on vessels.

For devices transmitting the information on the course over the ground (COG), the criteria are provided in [31,32]:−Transmission and definition errors should not exceed 0.2 degrees.−Statistical errors should be less than 1.0 degree.−The dynamic error amplitude should be less than 1.5 degrees.−The frequency of dynamic errors should be below 0.033 Hz and correspond to a period of not less than 30 s, if the dynamic error amplitude exceeds 0.5 degrees.

As regards gyrocompasses, the accuracy requirements are provided in [32], which established that at latitudes up to 60 degrees, the direction errors should not exceed the following values:
−0.25 degress · secant of the latitude—a residual fixed-state error after the correction for the impact of the velocity and heading, at the vessel’s velocity of up to 20 kn.−2 degrees—an error caused by a sudden change in velocity by 20 kn.−3 degrees—an error caused by a sudden change in the heading by 180 degrees, at a vessel’s velocity of 20 knots.−1 degree · secant of the latitude—errors of the transient and fixed state, caused by the swaying, pitching, yawing, simple harmonic motion during the period from 6 s to 15 s, on the maximum angles of 20 degrees, 10 degrees, and 5 degrees, respectively, and at the maximum horizontal acceleration not exceeding 1 m/s^2^.

For the above-mentioned requirements, a general accuracy assumption, characterized in Section 2.7, was made for three device types in terms of integrity.

### 2.3. Determination of the Vessel’s Reference Heading

The determination of error values and the degree of dependence of objects’ movement on the values of the errors occurring in compass device indications requires a comparison of compass indications with the reference direction. The variability of compass error values cannot be represented in a deterministic manner. One of the problems associated with the compass error determination is to determine the reference heading which is used to compare it with the heading read during the heading recordings being carried out. For research purposes, the reference heading was determined by comparing the readings of each compass with the reference rate determined by the POS MV OceanMaster Applanix Trimble hydrographic system (RMSE 0.02 degrees, 2 m baseline). Thanks to the hydrographic system, the ship keeps the course on the given hydrographic profile. The system is a component of a multi-beam sonar. Using such measurement technology, the accuracy of measurement can be higher than the accuracy offered by compasses. In addition, this technology is free of dynamic errors based on gyroscopic mechanical measurement elements.

Publications [18,33,34] compared the direction defined by the device with the reference heading to estimate the compass device indication accuracy. To this end, satellite systems were used to determine the heading and the indication accuracy for individual heading systems. Methods involving the comparison of the direction being defined with the hydrographic profile direction are commonly used to check the accuracy of compass indications when fixed errors occur. Despite the fact that these devices have a reduced accuracy that is determined by many factors, including the satellite visibility, wind speed, currents, the state of the sea, and the helmsman’s ability to maintain a steady heading on the hydrographic profile, this analysis ensures a comparison for shorter observation periods. Therefore, the methods described can only be used to determine the correction constant referred to as a compass error (CE). On the other hand, they are not applicable when testing the accuracy of devices under dynamic conditions. The measuring experiment performed the heading recording as a reading of the fixed movement direction indication using a classic gyrocompass with an internal correction system, a FOG compass, and a satellite compass. Recordings lasting from 200 to 350 s were excluded from further analysis due to insufficient data. In further tests, only three measuring series were used as the most valuable ones.

### 2.4. Determination of the Compass Heading Oscillation Amplitude

The primary task of the mathematical model for the vessel’s heading oscillation distribution is to determine the compass error amplitude. It is more convenient to study the complex spectrum, i.e., the vessel’s heading distribution and the vessel’s heading error oscillation distribution.

According to the relationship [5]:(1)$$HE\left(i\Delta t\right)=CH\left(i\Delta t\right)-RH\left(i\Delta t\right),$$
where:

*HE*(*i*Δ*t*)—heading error for the sample *i* = 0 … *N* − 1,

*RH*$\left(i\Delta t\right)$—reference heading for the sample *i* = 0 … *N* − 1,

*CH*(*i*∆*t*)—compass heading for the sample *i*.

The initial assumptions for the measuring campaign are primarily determined by the error values shown by classic and optical gyrocompasses. The long-term values of the errors occurring in compass indications are determined by the occurring dynamic deviation. Therefore, carrying out the analysis of compass error distribution in the time domain may appear insufficient. In this situation, the research methods of signal spectral analysis referred to as frequency domain analysis become particularly important.

### 2.5. Spectral Analysis of the Signal in the Frequency Domain

Based on the data processed in this manner, a spectral analysis in the frequency domain for heading errors from three compass systems was conducted.

For the compass indication accuracy assessment, a spectral analysis method was proposed, i.e., a frequency analysis known as Fourier transform presented in [5,7,8].

### 2.6. Band-Stop Model and Band-Pass Model of the FIR and IIR-Type Digital Signal Filtration

As regards digital (discrete) filtering, the theoretical analysis is based on the representation of signals in a discrete form. A digital filter processes the sequence of discrete sample values provided to its input. A digital filter may be a dedicated integrated circuit, a programmable processor, or a computer program.

For this reason, two forms of the discrete system description are possible, i.e., a system implementing an infinite impulse response (IIR) and a system implementing a finite impulse response (FIR).

The implementation of filters of this type is possible in the form of both a computer program and signal processes. In order to facilitate filter design, MATLAB was equipped with a graphic interface that enables easy design and analysis of FIR and IIR filters [35,36,37]. The amplitude spectrum of the recorded errors was divided into the useful signal band and the disturbance band. If the long-term errors of the three compasses correlate with each other and have the highest values for the 0.005–0.015 Hz frequency band, then this frequency range should be treated as noise. Considering that on the 270 and 90 degrees course, the velocity gyro error is zero [4], the values of long-term dynamic errors were considered as noise. If the error appeared only in the indications of a classic gyro-compass, then it should be assumed that the increase in the amplitude of dynamic errors in the frequency band 0.005–0.015 Hz should be treated as a useful signal. Such a situation takes place when the type two inertial error appears in the indications of gyro-compasses [4,8].

Based on the recorded data from three compasses and the amplitude spectrum of errors in the frequency domain, two models were determined, which are presented in Figure 1.

Model #1 assumes the use of a stop band in the range of 0.005–0.015 Hz.

Model #2 assumes the use of a passband in the range of 0.02 Hz.

By fulfilling the above conditions, the filter solution is achieved. Therefore, for the purposes of research work, it was proposed to apply an FIR filter in two variants and an IIR filter in two variants for the reduction in low-frequency heading oscillations.

Variant 1: FIR band-stop filter. Stop band of 0.005–0.015 Hz. Sampling frequency of 0.2 Hz.

Variant 2: FIR high-pass filter. Pass band of 0.02 Hz. Sampling frequency of 0.2 Hz.

Variant 3: IIR Butterworth band-stop filter. Stop band of 0.005–0.015 Hz. Sampling frequency of 0.2 Hz.

Variant 4: IIR Butterworth high-pass filter. Pass band of 0.02 Hz. Sampling frequency of 0.2 Hz.

When analyzing the nature of the occurring course errors in the frequency domain, the following assumption was made: if the amplitudes of the 3 compass errors do not correlate with each other, the deviation in the frequency domain will be a useful signal. A strong correlation of compass errors and the same trend of changes in compass errors indicate deviations caused by external factors—environmental error. The error in the frequency domain will be a noise.

Filter properties for two models and four variants are shown in the Table 2.

The methodology of testing the accuracy of compass indications involves the use of two models. Model #1 uses a stop band of 0.005–0.015 Hz. Model #2 uses a 0.02 Hz passband. The models use two types of filters: FIR and IIR, and two types of impulse response: band-stop and high-pass. Additionally, a Butterworth filter for the finite impulse response and an Equiripple filter for the infinite impulse response were used.

Equiripple as filter response type uses the Parks–McClellan method for estimation of order of filter which is a predictive approach towards the optimization of filters for minimization of order for a given set of frequencies [37]. The order of system needs to be kept as low as possible to reduce the number of delay elements, number of components during fabrication, and to reduce the mathematical complexity.

The Butterworth filter is characterized by a maximally flat amplitude frequency response. Compared to other filters of the same order, it exhibits the lowest amplitude drop in the transition band [37]. Filtering methods selected in this way allowed the elimination of noise based on the amplitude characteristics of long-time compass errors.

In the FIR filters, the response is the convolution function in the form of the sum of input signal samples with an impulse response.

The input argument is the Fourier transform and the frequency analysis of the signal spectrum, presenting the repeatability of the occurrence of heading errors during the vessel’s maneuvering on the pre-determined course. The number of points of which the impulse response is comprised was described as minimal. The stop-band in the band-stop filter and the pass-band in the high-pass filter were selected based on the spectral analysis of the signal in the frequency domain, for which the heading amplitude and the frequency of their occurrence indicates vessel steering errors made by the helmsman.

### 2.7. A Model for the Integrity of Information Derived from Compasses

In the methodology for testing compass accuracy, an important role is served by the reliability theory which performs operational characterization of the navigation system in the process of quantizing the state in which the particular system is currently found. The main aim of the study was to determine the integrity of the information derived from individual compasses based on the data from three tests. The study was carried out before and after the application of the methods filtering data derived from the compasses.

Integrity is the system’s ability to provide the user with timely warnings about the system’s inapplicability to the navigation process [38].

The duration of the correct operation of a compass with the required accuracy level is characterized as an exponential distribution of the time during which the system operated correctly (operating time) and of the time during which the system was faulty (failure time). If the process of information transfer by the system, resulting in the delivery of accurate information on the heading at any moment to the user, is a reliability process, then the system can be regarded as an information channel and tested using methods known from the reliability theory. With reference to the above paper, a model known from the Markov process theory used for the testing and description of technical objects in terms of integrity was proposed. A feature characteristic of the Markov chains is that the state of the system at the moment *n* + 1 is determined exclusively by the state *n*, and is not determined by the previous states of the system.

Based on the above assumption, for incoming data, a stochastic probability matrix of transitions between integrity states of the compass was determined, enabling the identification of the source of inaccurate data. The stochastic matrix presents the intensity of transitions between states.

The time value of the state of system integrity is a value that can be described using stochastic processes. Therefore, the author developed a model of compass integrity using the Markov chains methodology.

The stochastic process was denoted by the symbol:(2)$$\left\{S\left(t\right):t\in T\right\},$$

A special case of a stochastic process is a random sequence $\left\{S\left(n\right):n\;\epsilon \;\left(0,1,2,3,\ldots ,n\right)\right\}$ which is called a random chain. The values of random variables represent states of compass integrity in test. The nature of state changes can be assumed as Markov chain on a set of states:(3)$$S=\left\{S_{1},S_{2}\right\},$$
where:

*S*_1_*—*State *S(t)* = 0 means that in moment *t,*
$\sigma_{j}^{}>1\cdot \sec\mathrm{ant}\left(\varphi_{m}\right)\;\deg$, the system is in a failure state.

*S*_2_*—*State *S(t)* = 1 means that in moment *t,*
$\sigma_{j}^{}\leq 1\cdot \sec\mathrm{ant}\left(\varphi_{m}\right)\;\deg$, the system is in an operating state.

where:

$\sigma_{j}^{}$—difference of reference heading and compass heading j (deg);

$\varphi_{m}$—mean latitude (deg).

The moments when the system is in the failure state are the moments of the navigational structure renewal.

From the definition of the Markov chain, it is known that the Markov chain is characterized by the fact that the state at time n+1 depends only on the state at time n, and is independent of the state in the preceding moments.

The Markov chain is defined if the initial distribution:(4)$$P=\left[p_{ij}:i,j\in S\right],$$
and the matrix of transition probabilities:(5)$$P\left(X_{0}=i\right)=p_{i},i\in S,$$
where:(6)$$p_{ij}=P\left(X_{n+1}=j/X_{n}=i\right),\;n=0,\;1,\;2,\;3,\ldots N$$

n—number of samples of set N were given.

If n∈N, then the probability of transitions $p_{ij}\left(n-1,n\right)$ do not depend on n and Markov chain $\left\{X\left(n\right):n=0,1,2,3,\ldots m\right\}$ is the homogeneous Markov chain. In the case of homogeneous Markov processes, the probability of the process transition from state in the moment $t\left(n-1\right)$ to state in the moment $t\left(n\right)$ does not depend on previous moment $t\left(n-1\right)$ and after moment $t\left(n\right)$, but on a difference of moments $t\left(n\right)-t\left(n-1\right)$, so they depend on the time of transition. The transition probabilities can be written in the form of a square matrix with the number of rows/columns equal to the number of states.

Then, the stochastic matrix is defined as the probability of the intensity of transitions between the states of the system, i.e., a compass.

Then, pij denotes a transition from the state i ϵ S to the state j ϵ S for n+1.

In our case, the transition matrix takes the form of:(7)$$P=\left[\begin{array}{cc} p_{00} & p_{01}\\ p_{10} & p_{11} \end{array}\right]\;,$$

In addition, the initial distribution $p\left(0\right)=\left[0,\;1\right]$ was adopted for the study. This means that the system is in working state *S*_2_.

The limit probability was determined from the dependence:(8)$$\underset{n\to \infty }{\lim}p_{ij}\left(n\right)=\underset{n\to \infty }{\lim}P(X_{n+1}=j/X_{n}=i)=\underset{n\to \infty }{\lim}P(X_{n+1}=j)=\pi_{j},$$

To reach the intended goal, it was necessary to solve the set of linear equations:(9)$$\underset{i\in S}{\sum }\pi_{i}p_{ij}=\pi_{j},j\in S,$$
and
(10)$$\underset{i\in S}{\sum }\pi_{i}=1,$$
where:

*π*—limit probability

Probabilities determine the stationary distribution of homogenous Markov chain $\pi =\left[\begin{array}{cc} \pi_{0} & \pi_{1} \end{array}\right]$ with the matrix of transition probabilities:(11)$$P=p_{ij}:i,j\in S$$

Limit probabilities were calculated by solving a set of equations based on the matrix product:(12)$$\left[\begin{array}{cc} \pi_{0} & \pi_{1} \end{array}\right]\left[\begin{array}{cc} p_{00} & p_{01}\\ p_{10} & p_{11} \end{array}\right]=\left[\begin{array}{cc} \pi_{0} & \pi_{1} \end{array}\right],$$

As a result, a set of linear equations was obtained:(13)$$\left\{\begin{array}{c} p_{00}\pi_{0}+p_{10}\pi_{1}=\pi_{0}\\ p_{01}\pi_{0}+p_{11}\pi_{1}=\pi_{1}\\ \pi_{0}+\pi_{1}=1 \end{array}\right.\;,$$

On solving the set of equations, the limit probabilities *π*_0_, *π*_1_ were obtained.

## 3. Results

### 3.1. Preliminary Analysis of Recorded Data from Three Compass Systems

Data recordings from three heading systems and reference system are presented in Figure 2.

When analyzing the heading distribution in the time domain, presented in Figure 2, periodic oscillations caused mainly by the vessel’s yawing were observed. The differences in the amplitude and at the heading oscillation stage represent the diverse nature of the heading oscillation distribution resulting from the gyro element suppression of oscillation for the classic gyrocompass. The FURUNO SC50 satellite compass, equipped with MEMS (micro-electro mechanical systems) technology is more resistant to short-term rolling than the NAVIGAT 10 MK1 gyrocompass. The heading oscillations are determined by the type of functions estimating the vessel’s heading, such as the Kalman filter [39].

The examples of data recording present a perceptible similarity of oscillations in all indications of the three compasses. Of course, the results of the vessel’s yawing are particularly noticeable in Figure 2b,c. However, Figure 2c shows a gradual approaching of the values recorded by the NAVIGAT 10 MK1 to the values presented by other systems. This is a characteristic behavior of a classic gyrocompass immediately after a heading change.

In Figure 2b,c, the indications of the satellite compass and the FOG optical compass are analogous. The signal courses are in-phase with each other. As regards Figure 2a, a phase shift of the satellite compass indications, as compared to the reference system NAVIGAT 10 MK1, was observed. The differences are noticeable and amount to up to 3 degrees, while the accuracy of the devices is approximately 1 degree. During the experiment, the vessel maneuvered along the eastbound and westbound courses. Therefore, velocity deviation does not occur as an additional error in gyrocompass indications.

### 3.2. Results of the Spectral Analysis of Signals in the Frequency Domain

Three series of measurement data when applying the post-processing method were used in the testing of compass device accuracy. In addition, the effect of vessel’s movement and the accuracy of three heading systems were analyzed in terms of accuracy and the occurring dynamic errors of the devices. In further work on the methodology of compass device accuracy, stop-band filtration methods were applied to eliminate the above-mentioned errors. To this end, a spectral analysis of the signal was applied.

Based on the recorded heading oscillations, the heading error was determined according to relationship (3).

The values of heading oscillations—heading errors in the time domain with a value reduced by the POS MV Ocean Master Applanix Trimble—are provided in Figure 3.

Spectral analysis of the signal in the heading error frequency domain is determined by the long-term heading amplitude errors in the frequency domain. The distribution of satellite compass heading amplitude errors, in contrast to the FOG compasses and classic gyroscopic compasses, is similar, which indicates the devices’ susceptibility to errors. The values of these errors are determined by the object’s movement. The spectrum of the heading amplitude oscillations in the frequency domain are provided in Figure 4.

One-sided amplitude spectrum of the heading errors in the frequency domain for the three compasses is provided in Figure 4. The amplitude spectrum is presented for the harmonic component 0.015 Hz and for the maximum heading amplitude 1.3 deg. These values are determined by the long-term heading oscillations and the vessel’s inertial capabilities. Other results were obtained when analyzing Test #2, Figure 4b.

One-sided amplitude spectrum of the heading oscillations in the frequency domain represents the amplitude of 0.9 deg Figure 4b. As in the first measurement series, the greatest differences in the recorded heading in relation harmonic component 0.01 Hz of the heading errors, with the maximum spectrum to the reference heading were the reason for long-term heading errors due to steering errors made by the helmsman or environmental error as the effect of waving and constant heel on the gyrocompass readings. The swinging moment causes a precession of the X axis, thanks to which the X axis increases the azimuth in the west direction and the compass begins to indicate the direction with an error.

Similar results of testing for the vessel’s heading amplitude errors in the frequency domain were noted for Test #3. In this case, the harmonic component is 0.01 Hz, while the amplitude spectrum is 0.68 deg. Low-frequency errors below f=0.02 Hz are a phenomenon of unspecified heading fluctuations. The phenomenon is dictated by the occurring Schuler oscillations for the emerging heading errors for the harmonic component f=0.0018 Hz, which is shown in Figure 4a–c. The phenomenon causes an error in the indications of classic NAVIGAT 10 MK1 gyrocompass, where the deviation ranges from 0.7 deg to 1.3 deg. The amplitude deviation values result from the errors made by the helmsman during the vessel’s maneuvering on hydrographic profiles and the resulting environmental error as described above.

### 3.3. The Results of the Application of the Convolution Function in the form of FIR and IIR Filters in the Stop-Band and High-Pass Variants

One of the solutions used to reduce the heading amplitude errors for low frequencies is the application of the convolution function in the form of the sum of input signal samples with an impulse response. For the purposes of research work, four data filtration variants based on the FIR and IIR filter with a stop-band and the high-pass band were used. The results of the reduction in the heading amplitude errors based on the FIR or IIR filter used are provided in Figure 5, Figure 6, Figure 7 and Figure 8. Further on, mean compass heading error of recorded compass data and after the application of the FIR filter with a stop-band of 0.005–0.015 Hz, pass-band FIR 0.02 Hz, stop-band IIR 0.005–0.015 Hz, and pass-band IIR 0.02 Hz are shown in Table 3.

Variant 1: FIR band-stop filter. Stop-band of 0.005–0.015 Hz. Sampling frequency of 0.2 Hz (Figure 7).

Variant 2: FIR high-pass filter. Pass-band of 0.02 Hz. Sampling frequency of 0.2 Hz.

Variant 3: IIR Butterworth band-stop filter. Stop-band of 0.005–0.015 Hz. Sampling frequency of 0.2 Hz.

Variant 4: IIR Butterworth high-pass filter. Pass-band of 0.02 Hz. Sampling frequency of 0.2 Hz.

The FIR and IIR filters in the stop-band and high-pass variants were applied to suppress the heading amplitude errors in accordance with the assumptions presented for the four variants. The filtration was applied to suppress low frequencies of heading errors caused by helmsman errors. It is difficult to directly measure the frequency of heading errors caused by the helmsman, therefore the filter parameters are designated according to the computed accuracy of the device and based on the spectral analysis of the heading amplitude errors in the frequency domain. When applying the four proposed heading filtration variants, it is possible to filter out the harmonic components resulting from long-term heading errors caused by the helmsman’s steering errors and the inertial properties of the vessel. According to the test results presented in Figure 5, Figure 6, Figure 7 and Figure 8, the heading amplitude of the data filtered using the FIR and IIR filter in the stop-band and high-pass variants was reduced. Much better results of the filtration of heading amplitude errors were noted for the application of the high-pass variant with the pass-band of 0.02 Hz, as compared to the stop-band variant, for the suppression frequency range of 0.005–0.015 Hz. After comparing the results before and after the application of data filtration, the best reduction in the amplitude errors was achieved for the NAVIGAT FOG 3000 compass. The heading amplitude oscillations for this compass were reduced by four times by comparing the mean error values before and after the application of the FIR and IIR filters in four variants. In the other two compasses, a three-fold reduction in the heading error amplitude after the application of data filtration was noted.

### 3.4. The Results of Compass Integrity Testing

According to the assumptions provided in the methodology for testing the integrity of information derived from compass devices, the matrix of probabilities of the transition between the states of the system was determined based on the process of the intensity of transitions between the states, in accordance with Equations (12) and (13). When analyzing the intensity of transitions between the states in the Markov chain, the number of transitions between individual states in the process for Tests #1, #2, and #3 were determined (Table 4).

According to the adopted data processing criterion, it was assumed that each compass could be found in one of the two operational states, in the failure state, *S* = 0, and in the operating state of the system, *S* = 1. According to the assumption resulting from the quantisation of the operational state, and the accuracy criterion presented in Section 2.7, Table 4 provides the quantization results of the compass error depending on the system state in which the compasses (a) Navigat FOG 3000, (b) satellite compass Furuno SC50, and (c) NAVIGAT 10 MK1 were found. The integrity testing results apply to the data before and after the use of the convolution function, i.e., the data before processing and the data using four data filtration variants.

To this end, FIR and IIR filters in the band-stop and band-pass variants were applied. Table 4 show how many times the system changed its state of integrity depending on the data filtration variant applied, and for the three initial rows of recorded data. Different results were obtained for the FIR filter pass-band of 0.02 Hz, IIR band-stop of 0.005–0.015 Hz, IIR pass-band of 0.02 Hz, where the system, with the integrity criterion applied, was in the operating state for the entire operation period throughout the duration of the three tests. Large differences were noted for the FIR filter band-stop of 0.005–0.015 Hz. Better data filtering results were obtained when using the pass-band 0.02 Hz by the FIR and IIR filter.

According to Equation (9), the elements of the P matrix of the intensity of transitions between states for three compasses, two models FIR and IIR and four warrants and limit probabilities *π*(*i*) are presented in Table 5.

Table 5 lists the elements of the state transition intensity matrix. The elements of the matrix of the intensity of transitions between states determine the probability of the system compass in the working state or failure state, and the probability of transition between individual states. In accordance with the assumption, it was assumed that the system can be in one of two states. If the heading error is greater than 1.5 degrees, the system will be in the failure state. The last two columns (*π*(1), *π*(2)) are the limit probabilities that estimate the probability of the system being in the failure state and in the working state. It was observed that the passband from 0.02 Hz gives better results of heading filtration than the stop band 0.005–0.015 Hz. This is related to the appearance of the second type of inertial error in the final phase of maneuvering on a given hydrographic profile, which takes the maximum values from 3 to 5 degrees, within a quarter of the fluctuation period, i.e., after 20–22 min from the end of the heading alteration. Therefore, in the future, it is worth maneuvering on a given profile in accordance with the time that is a multiple of the Shuller period = 84.3 min.

The use of the 0.005–0.015 Hz stop band improves the filtration results more than 20%. That is, the integrity of the compass information is 20% higher in relation to the compass data, where filtration was not used. The use of the 0.02 Hz passband eliminates more than 95% of the indications from compass errors, including environmental error and type 2 inertial error observed in the NAVIGAT 10 MK1 indications.

Apart from the dynamic errors of the gyrocompass, there is also static deviation as a systematic error of the compass. The value of the systematic error after filtering the dynamic errors with the FIR, IIR filter is from 0.02 to 0.38 degrees. These values can be observed on the basis of the analysis of the amplitude spectrum of compass errors (Figure 4). The value of this error is influenced by the deviation of the sensitivity of the tracking system, environmental deviation (waving), deviation of the compass assembly, and instrumental compass error. Details can be found in [4].

## 4. Discussion

The article presents a novel method for validating compass devices based on the previously recorded vessel’s heading derived from three compass devices, i.e., an analytic compass with a closed fiber optic loop, a satellite compass, and a classic gyroscopic compass with an internal correction system, installed on a platform, i.e., a hydrographic vessel during hydrographic operations performed. After initial filtering of the data, a spectral analysis of the recorded heading as a function of frequency was conducted. To this end, fast Fourier transform was applied. A synthetic summary of the results of spectral analysis in the frequency domain for the errors provided the basis for performing further research work related to filtering out of the heading errors due to the helmsman’s errors and the wind impact resulting in the vessel’s drift. For the purposes of research work, convolution functions in the form of the sum of input signal samples with a response, i.e., the filter with a finite impulse response (FIR) and an infinite impulse response (IIR), were applied to compare the effectiveness of methods estimating the vessel’s heading. A total of four variants were applied: a band-stop model and band-pass model of the FIR and IIR-type digital signal filtration with the stop-band of 0.005–0.015 Hz or the pass-band of 0.02 Hz and the sampling frequency of 0.2 Hz. The input argument was the Fourier transform and the frequency analysis of the signal spectrum, presenting the repeatability of the occurrence of heading errors during the vessel’s maneuvering on the pre-determined course at the time of the marine hydrographic sounding performance by a survey ship. Further study applied a model for the integrity of information derived from compasses. To this end, the method known from reliability theory was used; a state- and time-discrete Markov chain, performing operational characterization of the navigation system in the process of quantizing of the state in which the particular system is currently found. The integrity of the system was estimated before and after the use of the four variants filtering the data of the heading error amplitude spectrum. When analyzing the intensity of transitions between states in the Markov chain, the number of transitions between individual states in the process for Tests #1, #2, and #3 was determined. When performing the system state quantization, slightly better filtration results were observed for the IIR filter as compared to the FIR filter in the stop-band of 0.005–0.015 Hz. Therefore, a minimum drop in the intensity of transitions from the system failure state, when using the IIR filter, and changes from the stop-band of 0.005–0.015 Hz to a band of 0–0.02 Hz were observed.

The mean error of the compass heading following the use of FIR and IIR filters was reduced from 20 to 100% for compass depending on the test number and compass type. The use of an excessively wide stop-band may result in filtering out the data that result from dynamic errors of classic gyrocompasses.

## 5. Conclusions

This paper answers the four research questions presented at the start.

The results of integrity analyses for information derived from three compass systems proved the effectiveness of the solution involving the application of methods based on testing the amplitude spectrum of the vessel’s heading oscillations. The FIR and IIR filters applied effectively eliminated low-frequency oscillations of the compass heading for three compass systems, which is proven by the integrity testing results. By using a state- and time-discrete Markov chain, the results of the intensity of transitions between states of system service ability and integrity, i.e., the three naval compasses, were compared.

As regards the compass heading amplitude oscillations caused by the vessel steering errors and the environmental factors such as a drift caused by the wind impact on the vessel, a stop-band of 0.005–0.015 Hz and a high-pass band starting from 0.020 Hz were applied, which ensured good results of data filtration for the use of the FIR and IIR filter.

In order to determine the intensity of transitions between operational states of the system and to detect the differences in the accuracy of compass devices before and after the application of the FIR and IIR in four variants, it was assumed that the system was in a serviceable state when the heading amplitude oscillations did not exceed $1\;\deg\cdot \sec\mathrm{ant}\;\left(\varphi_{m}\right)$. This assumption arises from the fact that the dynamic error amplitude should be less than 1.5 degrees [31,32]. In addition, the errors of the transient and fixed state (caused by the pitching, yawing, simple harmonic motion during the period from 6 s to 15 s, on the maximum angles of 20 degrees, 10 degrees, 5 degrees, respectively, and the maximum horizontal acceleration not exceeding 1 m/s^2^) should not exceed 1 degree · secant of the latitude [32].

The presented method for testing the correctness of compass device indications using the spectral analysis of heading oscillations can be followed based on the previously recorded data derived from compasses (post-processing). However, the question should be asked: is it possible nowadays to automate the compass device validation process based on the FFT method and the amplitude spectrum of heading oscillations? This integrity testing method in real-time would undoubtedly affect the quality of the operation of devices performing the tasks of automated heading steering, where adaptation autopilots are already used based on artificial intelligence and EKF filtration in real-time. Moreover, as regards the information derived from the AIS system, dynamic information in the AIS positioning reports is, in many cases, devoid of data on the gyrocompass heading. The heading estimation with the backward correction, when applying the heading spectral analysis with several seconds of sampling could, to a certain extent, eliminate the limitations of the AIS system information unsuitability. In this situation, information on the gyroscope heading could be obtained in real-time at any moment. In the future, in such a study, it would be worth considering the application of Markov processes of the second order. In the Markov model, second order elementary states correspond to sequences of two nodes, and thus can be identified with directed edges in the original network. The state space of the model defines a new network describing the observed dynamics or memory network. The structural properties of the memory network can now be studied by many tools of network science, allowing one to uncover interesting patterns of flow in a second order Markov process [40].

## Figures and Tables

**Figure 1 sensors-22-02530-f001:**
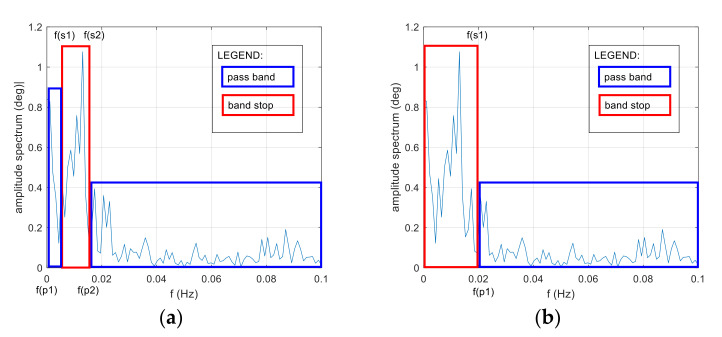
General model oscillation, dynamic conditions. (**a**) Stop band of 0.005–0.015 Hz. (**b**) Pass band of 0.02 Hz. f(p)—frequency pass band, f(s)—frequency stop band.

**Figure 2 sensors-22-02530-f002:**
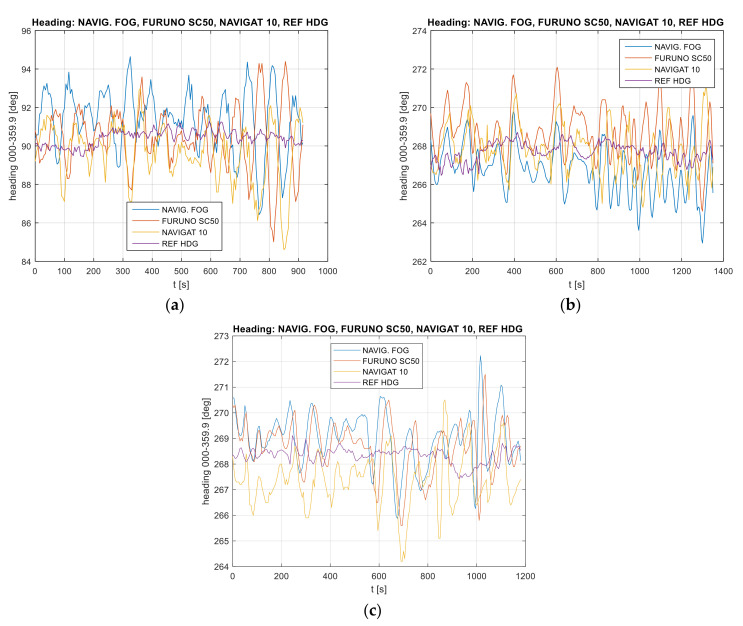
Heading recordings: (**a**) Test #1, (**b**) Test #2, (**c**) Test #3. The measurement was performed by the post-processing method.

**Figure 3 sensors-22-02530-f003:**
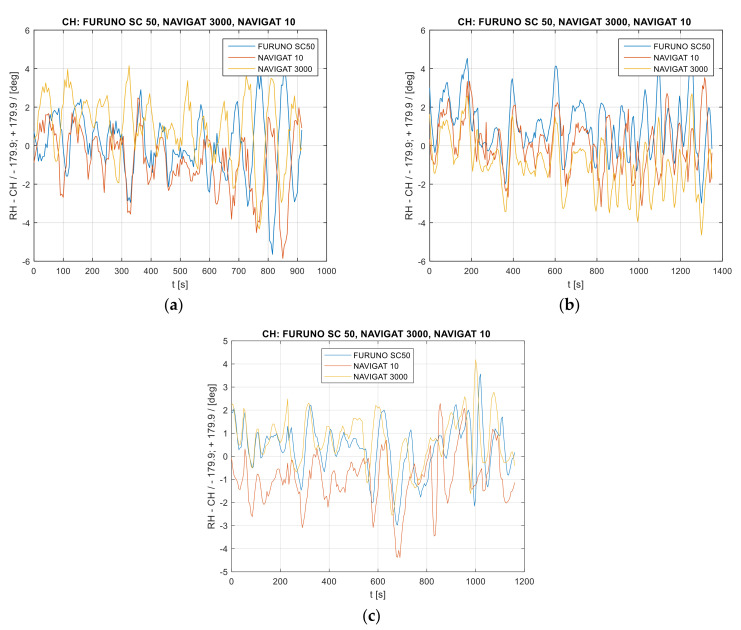
Heading recordings: (**a**) Test #1, (**b**) Test #2, (**c**) Test #3. The heading error amplitude distribution in the time domain. The heading values were reduced by the mean measurement from three heading systems.

**Figure 4 sensors-22-02530-f004:**
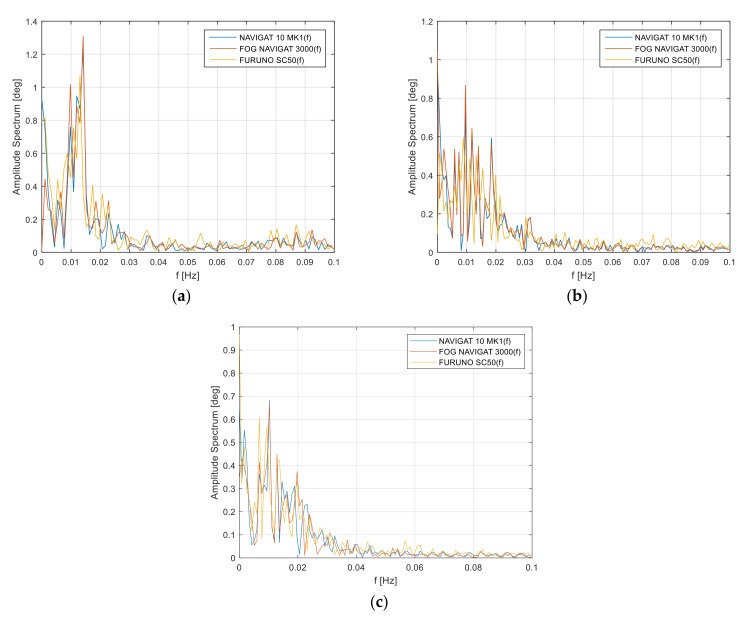
One-sided amplitude spectrum of the heading errors in the frequency domain: (**a**) Test #1, (**b**) Test #2, (**c**) Test #3.

**Figure 5 sensors-22-02530-f005:**
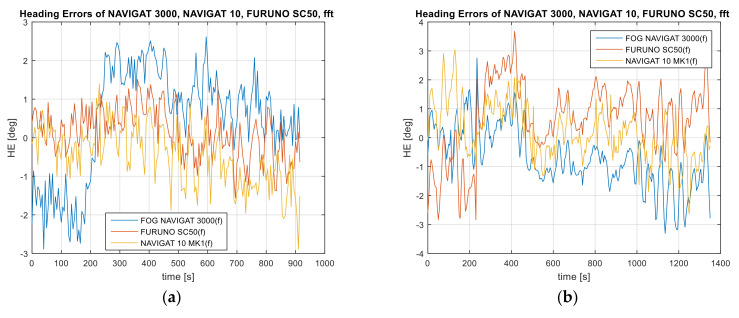
Heading amplitude error of the FOG 3000, FURUNO SC50, and NAVIGAT 10 MK1 compass heading after the application of the FIR filter with a stop-band of 0.005–0.015 Hz, (**a**) Test #1, (**b**) Test #2, (**c**) Test #3.

**Figure 6 sensors-22-02530-f006:**
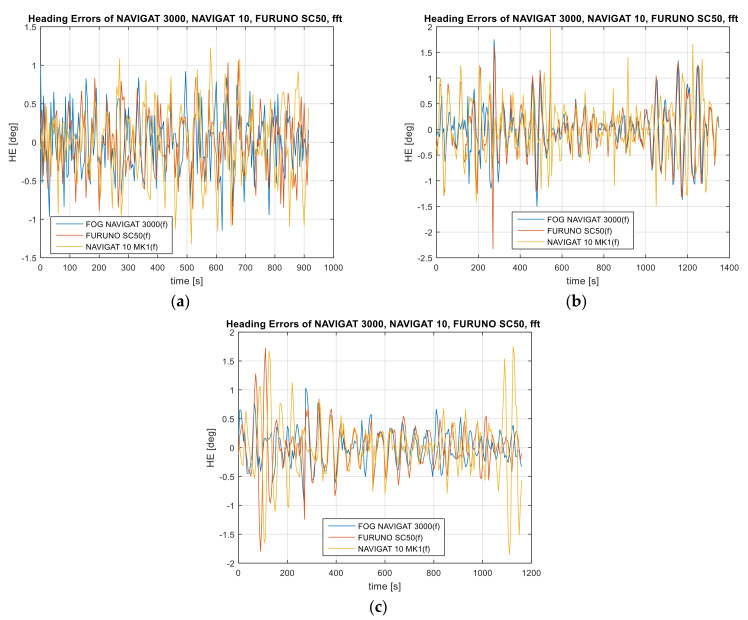
Absolute errors of the FOG 3000, FURUNO SC50, and NAVIGAT X MK1 compass heading after the application of the FIR filter with a pass-band of 0.02 Hz, (**a**) Test #1, (**b**) Test #2, (**c**) Test #3.

**Figure 7 sensors-22-02530-f007:**
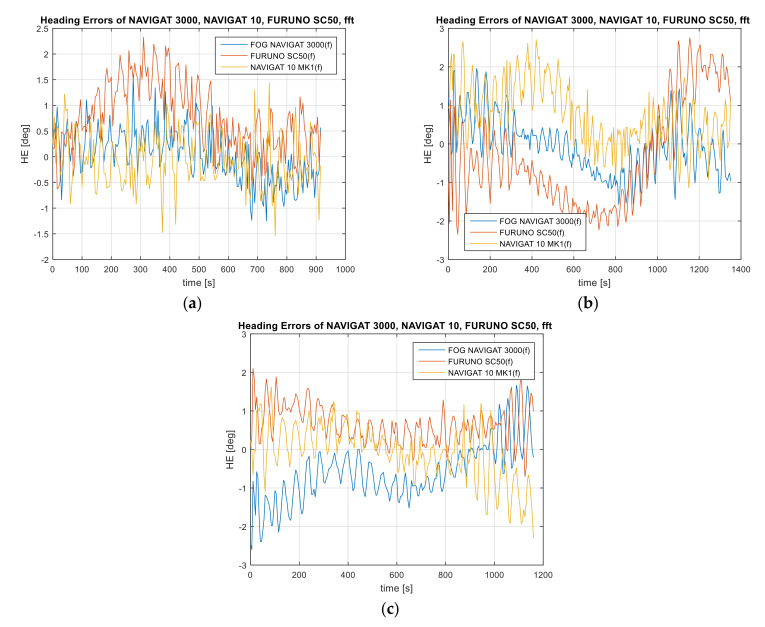
Absolute errors of the FOG 3000, FURUNO SC50, and NAVIGAT 10 MK1 compass heading after the application of the IIR filter with a stop-band of 0.005–0.015 Hz, (**a**) Test #1, (**b**) Test #2, (**c**) Test #3.

**Figure 8 sensors-22-02530-f008:**
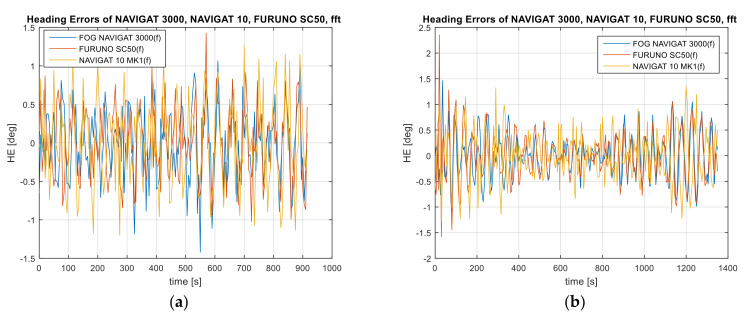
Absolute errors of the FOG 3000, FURUNO SC50, and NAVIGAT X MK1 compass heading after the application of the IIR filter with a pass-band of 0.02 Hz, (**a**) Test #1, (**b**) Test #2, (**c**) Test #3.

**Table 1 sensors-22-02530-t001:** The accuracy and mean error of the compasses, taking into account the latitude of 54.6 degrees N.

Compass Type	Accuracy (Degree)	Mean Error (Degree)
NAVIGAT FOG 3000	0.4 deg ∙ s (*φ*_*m*_)	0.7
FURUNO SC50	0.5	0.5
NAVIGAT 10 MK1	0.4 deg ∙ s (*φ*_*m*_)	0.7
POS MV Ocean Master Applanix Trimble	0.02	0.02

**Table 2 sensors-22-02530-t002:** Filter properties for the band-stop model 0.005–0.15 Hz and pass-band model 0.02 Hz.

Model	Variant	Filter Type	Resp. Type	Design Method	F(s) (Hz)	Fpass1 (Hz)	Fstop1 (Hz)	Fstop2 (Hz)	Fpass2 (Hz)
#1	#1	FIR	bandstop	Equripple	0.2	0.005	0.01	0.015	0.02
#2	#2	FIR	high pass		0.2	-	0	0.015	0.02
#1	#3	IIR	bandstop	Butterworth	0.2	0.002	0.005	0.015	0.02
#2	#4	IIR	high pass		0.2	-	0	0.015	0.02

F(s)—sampling frequency. Fpass1—first passband frequency. Fstop1—first stopband frequency. Fstop2—second stopband frequency. Fpass2—second passband frequency.

**Table 3 sensors-22-02530-t003:** Mean compass heading error of recorded compass data and after the application of the FIR filter with a stop-band of 0.005–0.015 Hz, pass-band FIR 0.02 Hz, stop-band IIR 0.005–0.015 Hz, pass-band IIR 0.02 Hz.

Compass Type	Recorded Compass Data (deg)	Stop-Band FIR 5–15 mHz (deg)	Pass-Band FIR 20 mHz (deg)	Stop-Band IIR 5–15 mHz (deg)	Pass-Band IIR 20 mHz (deg)
NAVIGAT FOG 3000	1.31	1.01	0.31	0.59	0.32
FURUNO SC 50	1.20	0.86	0.34	0.92	0.35
NAVIGAT 10 MK1	1.22	0.94	0.39	0.62	0.40

**Table 4 sensors-22-02530-t004:** Quantization results of the compass error depending on the system state, all iterations of Tests: #1, #2, #3.

Filter Type	Frequency Band-Stop (mHz)	Compass Type	Test No	*S*(0)→*S*(0)	*S*(0)→*S*(1)	*S*(1)→*S*(0)	*S*(1)→*S*(1)	Number of States
-	-	FOG	1, 2, 3	180	52	50	406	688
-	-	SC	1, 2, 3	151	43	40	454	688
-	-	MK1	1, 2, 3	137	57	54	440	688
FIR	5–15	FOG	1, 2, 3	61	40	38	549	688
FIR	5–15	SC	1, 2, 3	87	27	25	549	688
FIR	5–15	MK1	1, 2, 3	59	43	41	545	688
IIR	5–15	FOG	1, 2, 3	6	12	9	661	688
IIR	5–15	SC	1, 2, 3	120	43	40	485	688
IIR	5–15	MK1	1, 2, 3	32	21	21	612	688
FIR	0–20	FOG	1, 2, 3	0	5	2	681	688
FIR	0–20	SC	1, 2, 3	1	6	3	678	688
FIR	0–20	MK1	1, 2, 3	4	11	8	665	688
IIR	0–20	FOG	1, 2, 3	0	3	0	685	688
IIR	0–20	SC	1, 2, 3	0	5	2	681	688
IIR	0–20	MK1	1, 2, 3	0	3	0	685	688

**Table 5 sensors-22-02530-t005:** The elements of transition matrix P from the state *i* ϵ *S* to the state *j* ϵ *S* for *n* + 1 and limit probabilities *π*(*i*), for all iterations of Test #1 #2 #3.

Filter Type	Frequency Band-Stop (mHz)	Compass Type	Test No	*p*(00)	*p*(01)	*p*(10)	*p*(11)	*π*(1)	*π*(2)
-	-	FOG	1, 2, 3	0.780	0.220	0.114	0.886	0.340	0.660
-	-	SC	1, 2, 3	0.748	0.252	0.088	0.912	0.277	0.723
-	-	MK1	1, 2, 3	0.694	0.305	0.118	0.882	0.292	0.707
FIR	5–15	FOG	1, 2, 3	0.576	0.427	0.072	0.928	0.151	0.849
FIR	5–15	SC	1, 2, 3	0.476	0.524	0.045	0.955	0.144	0.856
FIR	5–15	MK1	1, 2, 3	0.523	0.476	0.072	0.928	0.147	0.853
IIR	5–15	FOG	1, 2, 3	0.250	0.750	0.012	0.987	0.024	0.976
IIR	5–15	SC	1, 2, 3	0.630	0.370	0.081	0.918	0.224	0.776
IIR	5–15	MK1	1, 2, 3	0.324	0.676	0.031	0.969	0.069	0.930
FIR	0–20	FOG	1, 2, 3	0.000	1.000	0.002	0.998	0.007	0.993
FIR	0–20	SC	1, 2, 3	0.111	0.889	0.004	0.996	0.009	0.991
FIR	0–20	MK1	1, 2, 3	0.121	0.879	0.011	0.989	0.021	0.979
IIR	0–20	FOG	1, 2, 3	0.000	1.000	0.000	1.000	0.004	0.996
IIR	0–20	SC	1, 2, 3	0.000	1.000	0.002	0.998	0.007	0.993
IIR	0–20	MK1	1, 2, 3	0.000	1.000	0.000	1.000	0.004	0.996

## Data Availability

Not applicable.

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
