# Peer review of "Methodology for Verifying the Indication Correctness of a Vessel Compass Based on the Spectral Analysis of Heading Errors and Reliability Theory"

_sensors, 2022, doi:10.3390/s22072530_

Round 1
Reviewer 1 Report
The article Methodology for Verifying the Indication Correctness of a Vessel Compass Based on the Spectral Analysis of Heading Errors and Reliability Theory considers a compass's errors and suggests a way to check if it is reliable, based on the Markov processes approach. I think that the article fits the journal's scope, includes valuable results, and has the potential to be published in Sensors. However, I have several issues that the authors should meet before.
1) I would not recommend excessive self-citation like 4–11.
2) I am not sure that a scientific article should include well-known issues like Fourier Transform or FFT. Moreover, I do not know why the author used 4 references for Fourier Transform (4) [13,16,38,39]. The same is for 2.8. Probably, some parts are essential for the article, but the article should not be a textbook.
3) The article is difficult to read. The text needs significant rewriting and structuring. I would recommend decreasing the introduction but clearly stating the task (what are you planning to do, what do you compare).
4) There are too many tables in figures in the main text. I would recommend shifting them in the supplementary and showing the main features in the article because it is difficult for a reader to analyze them.
5) It is not clear how systematic errors could influence the results.
Minor remarks:
1) “The state of the sea of 1-2, the wind affected the vessel from the SW-2B (Beaufort)” - unclear
2) Figure 1 is useful even outside the article, but I would recommend increasing fonts and making lines more contrasted. The same is for other figures.
3) Figures 2 and 3 should be remade like graphs (or even one graph) – see fig. 1.
4) Figures 11-13 are so small that I barely see anything.
5) Fill in the “Data Availability Statement:” and “Acknowledgments:”. Now they are from the template.
Author Response
REVIEWER 1:
Comments and Suggestions for Authors
ISSUE_1: I would not recommend excessive self-citation like 4–11.
ANSWER_1: Citations 4-11 have been deleted.
ISSUE_2: I am not sure that a scientific article should include well-known issues like Fourier Transform or FFT. Moreover, I do not know why the author used 4 references for Fourier Transform (4) [13,16,38,39]. The same is for 2.8. Probably, some parts are essential for the article, but the article should not be a textbook.
ANSWER_2: I deleted citations 16, 38, 39 in the manuscript. I corrected Chapter 2 and removed some of the information in it.
ISSUE_3: The article is difficult to read. The text needs significant rewriting and structuring. I would recommend decreasing the introduction but clearly stating the task (what are you planning to do, what do you compare).
ANSWER_3: I have reduced the content of the introduction. The description of the content of the article and the problem under consideration is included in the introduction. In my opinion, after the corrections, I clearly presented what I will do in the research. The methodology is similar to that contained in [1,2]:
[1] Jaskólski, K.; Felski, A.; Piskur, P. The Compass Error Comparison of an Onboard Standard Gyrocompass, Fiber-Optic Gyrocompass (FOG) and Satellite Compass. Sensors 2019, 19, 1942.
and
[2] Felski, A.; Jaskólski, K.; Zwolak, K.; Piskur, P. Analysis of Satellite Compass Error’s Spectrum. Sensors 2020, 20, 4067.
The difference in the methodology contained in the presented manuscript is that in this article I used the reference rate from the hydrographic system and not as in the publication [1] and [2] where I set the average rate as the reference rate. Using the FFT, I analyzed the amplitude spectrum of compass errors and based on the occurring harmonic components of the signal at low frequencies, I applied FIR and IIR filtering, cutting out the band with harmonics that appeared at low frequencies.
Additionally, I analyzed the integrity of 3 compasses. I applied Markov processes discreet in states and in time - Markov Chain.
ISSUE_4: There are too many tables in figures in the main text. I would recommend shifting them in the supplementary and showing the main features in the article because it is difficult for a reader to analyze them.
ANSWER_4: I reduced the number of tables in the manuscript. The tables with the results from each measurement campaign are presented in individual tables, which gives a general idea of the integrity and accuracy of compasses.
ISSUE_5: It is not clear how systematic errors could influence the results.
ANSWER_5: Apart from the dynamic errors of the gyrocompass, there is also static deviation as a systematic error of the compass. The value of the systematic error after filtering the dynamic errors with the FIR, IIR filter is from 0.02 to 0.3 deg. These values can be observed on the basis of the analysis of the amplitude spectrum in the frequency domain of the occurring compass errors. The value of this error is influenced by the deviation of the sensitivity of the tracking system, environmental (waving) deviation, the deviation of the compass mounting, the instrumental error of the compass.
Minor remarks:
ISSUE_1:“The state of the sea of 1-2, the wind affected the vessel from the SW-2B (Beaufort)” – unclear
ANSWER_1: These are meteorological data that are important in determining the situation in which there is (environmental) wave deviation in the gyrocompasses. It is a static, systematic error. The (environmental) wave deviation depends on the direction of the wave's attack on the hull and is the greatest for waves from directions with a course angle of 045, 135, 225, 315 degrees. Systematic error values according to the FFT figure take values from 0.02 to 0.5 deg.
ISSUE_2: Figure 1 is useful even outside the article, but I would recommend increasing fonts and making lines more contrasted. The same is for other figures.
ANSWER: I corrected Figure 1 and enlarged the remaining Figures. They are much clearer. I use MATLAB and these are the original software colors..
- Figures 2 and 3 should be remade like graphs (or even one graph) – see fig. 1.
ANSWER: I removed Fig 2,3. Data describing the FIR, IIR models are included in Table 2. These will be the same figures as in Figure 1.
- Figures 11-13 are so small that I barely see anything.
ANSWER: The data contained in these figures are presented in the tables. I removed Figures as recommended by another Reviewer.
- Fill in the “Data Availability Statement:” and “Acknowledgments:”. Now they are from the template.
ANSWER: Thank you for your valuable attention. After reducing the number of figures and data in the tables in the manuscript, I remain in the current version. If the Editor decides that I should transfer Figures to Statement, I will respond to the recommendation..
Reviewer 2 Report
The author presents a novel method to compare the indication correctness of different compass types. In this study the performance of a satellite-based, classic gyrocompass and a FOG compass are compared. Therefore three different datasets are collected.
The method is based on a spectral analysis (FFT) of the observations of the different compass types. Based on the FFT, steering and drift errors of the vessel are identified and removed using different filter setups. Within four different filter configurations, both, filters with finite and infinite impulse response are covered. In order to compare the performance of the different filter setups, a discrete Markov process is modeled and applied to each filter setup.
The paper itself is divided into five sections: Introduction, Materials and Methods, Results, Discussion and ends with a Conclusion. There are minor mistakes in grammar and punctuation (e.g. after inserted cites). Some Sections could provide additional references (e.g. Sec. 2.4, 2.5, 2.6).
The paper appears unnecessarily long which makes it difficult to read. The paper should be restructured and shortened accordingly. this concerns especially, the introduction and Section 2, covering basic methods like FFT, IIR, and FIR.
The MATLAB Figures 2 and 3 are very small and difficult to read. Since all the information is already given in Table 2, are those Figures necessary?
The same applies for Figures 11-13. They plots are very small and the information is already contained in previous tables.
Table 3-7 could include an average over all tests.
Author Response
REVIEWER 2
Comments and Suggestions for Authors
ISSUE_1: There are minor mistakes in grammar and punctuation (e.g. after inserted cites). Some Sections could provide additional references (e.g. Sec. 2.4, 2.5, 2.6).
ANS_1: I have corrected the errors as far as I can. I deleted much of the text from Sec. 2.4, 2.5, 2.6. The methodology of research works is included in many of my publications.
ISSUE_2: The paper appears unnecessarily long which makes it difficult to read. The paper should be restructured and shortened accordingly. this concerns especially, the introduction and Section 2, covering basic methods like FFT, IIR, and FIR.
ANS_2: I reduced the content of introduction and removed some content from Section 2 covering basic methods like FFT, IIR, and FIR.
ISSUE_4: The MATLAB Figures 2 and 3 are very small and difficult to read. Since all the information is already given in Table 2, are those Figures necessary?
The same applies for Figures 11-13. They plots are very small and the information is already contained in previous tables.
ANS_4: I removed redundant Figures and changed the tables according to the reviewer's suggestions. On the basis of all the measurement results from 3 tests, I determined the intensity of being in operating states in one table.
ISSUE_5: Table 3-7 could include an average over all tests.
ANS_5: Based on the contents of tables 3-7, I have compiled one table with the results.
Thank You for Your valuable comments, Best regards.
Round 2
Reviewer 1 Report
The authors improved their article to exclude well-known information. From my viewpoint, now the article is clear and correct, the results obtained are useful for engineers, especially results on compasses' problems. So I think the article can be accepted for publication.